# Development of Carbazole Derivatives Compounds against *Candida albicans*: Candidates to Prevent Hyphal Formation via the Ras1-MAPK Pathway

**DOI:** 10.3390/jof7090688

**Published:** 2021-08-25

**Authors:** Young-Kwang Park, Jisoo Shin, Hee-Yoon Lee, Hag-Dong Kim, Joon Kim

**Affiliations:** 1Laboratory of Biochemistry, Division of Life Sciences, Korea University, Seoul 02841, Korea; 01091210524@korea.ac.kr; 2Department of Chemistry, KAIST, Daejeon 34141, Korea; lucky5624@kaist.ac.kr (J.S.); leehy@kaist.ac.kr (H.-Y.L.); 3HAEL, TechnoComplex Building 603-3, Korea University, Seoul 02841, Korea; kste99@yahoo.co.kr

**Keywords:** fungi, morphogenesis, *Candida albicans*, pathogenicity, Ras1, MAPK pathway, drug resistance, biofilm formation, candidiasis, drug development

## Abstract

Morphogenesis contributes to the virulence of the opportunistic human fungal pathogen *Candida albicans*. Ras1-MAPK pathways play a critical role in the virulence of *C. albicans* by regulating cell growth, morphogenesis, and biofilm formation. Ume6 acts as a transcription factor, and Nrg1 is a transcriptional repressor for the expression of hyphal-specific genes in morphogenesis. Azoles or echinocandin drugs have been extensively prescribed for *C. albicans* infections, which has led to the development of drug-resistant strains. Therefore, it is necessary to develop new molecules to effectively treat fungal infections. Here, we showed that Molecule B and Molecule C, which contained a carbazole structure, attenuated the pathogenicity of *C. albicans* through inhibition of the Ras1/MAPK pathway. We found that Molecule B and Molecule C inhibit morphogenesis through repressing protein and RNA levels of Ras/MAPK-related genes, including *UME6* and *NRG1*. Furthermore, we determined the antifungal effects of Molecule B and Molecule C in vivo using a candidiasis murine model. We anticipate our findings are that Molecule B and Molecule C, which inhibits the Ras1/MAPK pathway, are promising compounds for the development of new antifungal agents for the treatment of systemic candidiasis and possibly for other fungal diseases.

## 1. Introduction

*Candida albicans* is a commensal fungal pathogen that causes opportunistic infections in humans. Life-threatening systemic fungal infections occur mainly in immunocompetent and immunocompromised humans, such as patients with AIDS, immunosuppressant-treated organ transplant patients, and recently patients with COVID-19 [1,2,3]. *Candida* infection has shown an increasing prevalence and severity with an increase in AIDS prevalence, chemotherapy, and organ transplantation. Candidemia accounts for approximately 9% of nosocomial bloodstream infections, and when *C. albicans* invades the internal organs, it can cause severe candidiasis with a mortality rate of approximately 40%.

Morphogenesis is critical for the pathogenicity of *C. albicans* [4,5]. It has two main forms: yeast and hyphal. The dimorphism that allows the ability to switch between yeast and hyphal growth forms is essential for pathogenicity. The yeast form is required for adhesion and colonization of the epithelial cells of the host, while the hyphal form is required for the penetration and invasion of the endothelial cells of the host [6]. Morphogenesis has various environmental cues, such as nutrient limitation, alkaline pH, quorum sensing molecules, serum, elevated temperature, and elevated CO_2_ [7,8]. For in vitro experiments, serum exposed to elevated temperature (37 °C) was used for hyphal formation [9]. Therefore, novel drugs that have the ability to inhibit the morphogenesis of *C. albicans* must be screened. In *C. albicans*, MAPK and Ras/PKA-dependent pathways regulate nutrient sensing and acquisition, stress response, and pathogenesis, as well as play a potent role in hyphal formation [10,11]. The small GTPase Ras1 in this fungus affects both pathways and acts as a switch for hyphal formation signals, such as serum, elevated temperature, and nutrient limitation [11]. As one of the MAPKs, Cek1 has a role in biofilm formation and filamentous growth in *C. albicans*, while another MAPK, Mkc1, has a function in the formation and maintenance of cell wall structure [10]. CaUme6 is a transcription factor that maintains filamentous growth in hyphal formation and upregulates hyphal-specific genes (HSGs) encoding hyphal cell-related components, such as Hwp1, Als3, and Ece1 proteins [12]. The transcriptional repressor Nrg1 works with Ume6 as a negative feedback loop to control the expression of HSGs in hyphal growth environmental signals [13,14].

Fluconazole, an azole drug typically used as an antifungal agent, targets the enzyme essential for cell wall ergosterol biosynthesis. Other antifungal drugs, such as amphotericin B (polyene) and echinocandin target the fungal membrane and cell wall synthesis, respectively. In the ergosterol biosynthesis pathway, azoles inhibit ERG11, which encodes 14α-demethylase (cytochrome P450 enzyme lanosterol demethylase) [15]. The azole-resistant *C. albicans* strains have been isolated from patients prescribed chemotherapy drugs. The azole-resistant *C. albicans* strains have some features, such as point mutations and the overexpression of ERG11, azole target, and overexpression of drug efflux pumps Cdr1 and Mdr1. To effectively treat infections caused by these drug-resistant *C. albicans* strains, the necessity of new antifungal drugs that have different targets from those of the commonly used drugs is emphasized.

Carbazole derivatives, which are widely used because of their biological activity, are receiving increasing attention in antitumor, antibacterial, and antifungal studies [16,17,18]. They have been shown to exhibit antifungal activity against a variety of fungi, including *C. albicans*, but the exact underlying mechanism has not been studied. In our previous study, we investigated the correlation between vacuolar activity and pathogenicity in *C. albicans* through deletion of V-ATPase subunits vma4 and vma10 [19], and the results suggested that targeting vacuoles with novel fungicidal agents would be beneficial. In this study, 31 compounds containing a carbazole structure were synthesized as a result of preparing an analog, such as bafilomycin A1 with vacuole inhibitory activity. Molecule B and Molecule C were selected through the screening and evaluation of 31 carbazole derivatives. Both showed excellent efficacy in inhibiting morphogenesis and pathogenicity but did not significantly affect the vacuolar activity in *C. albicans*. Here, we describe Molecule B and Molecule C, which inhibit the Ras1/MAPK pathway to prevent hyphal morphogenesis and consequently attenuate the pathogenicity in *C. albicans*. Overall, this study focused on elucidating the mechanism of action and evaluating the antifungal activity using both in vitro and in vivo models to confirm the potential of Molecule B and Molecule C as novel antifungal drugs for *Candida* infection.

## 2. Materials and Methods

### 2.1. Strains, Culture Media, and Chemicals

The *C. albicans* strains, including fluconazole and caspofungin-resistant strains, primers, and 31 carbazole derivatives used in this study are listed in Table 1, Appendix A, respectively. To generate the Ume6-myc and Nrg1-myc strains, PCR-based gene disruption was performed using HIS1 and 9myc-NAT1 cassettes. The primers for amplification of the cassettes (HIS1 and 9myc-NAT1) are listed in Appendix A. To generate the UME6-9myc and NRG1-9myc strains, pFA6a-9myc-NAT1 was used for UME6-9myc-NAT1 and NRG1-9myc-NAT1 cassettes. These DNA cassettes were transformed into the *ume6::HIS1/UME6* and *nrg1::HIS1/NRG1* strains. To confirm strain generation, genomic DNA was isolated, and PCR and western blotting were performed with specific primers and antibodies.

*Saccharomyces cerevisiae* (Appendix A) and *C. albicans* cells were cultivated at 30 °C in YPD medium (1% yeast extract, 2% peptone, and 2% dextrose), as previously described [20,21]. For the induction of hyphal morphogenesis in *C. albicans*, incubation was performed in YPD medium containing 10% fetal bovine serum (FBS) at 37 °C with shaking. A complete synthetic medium (0.76% yeast nitrogen base without amino acids and 2% dextrose) supplemented with the appropriate auxotrophic requirements and uridine (50 µg/mL), was used to select positive transformants. RPMI 1640 (pH 7.0) with 0.165 M MOPS liquid medium was used for susceptibility determination according to the Clinical and Laboratory Standards Institute (CLSI) document M27-A3 [22].

Caspofungin and fluconazole were dissolved in DMSO to a final concentration of 20 mg/mL. All stock solutions of 31 carbazole derivatives were dissolved in sterile distilled water or methanol to a final concentration of 20 mg/mL. All stock solutions were stored at −20 °C until use.

**Table 1 jof-07-00688-t001:** *Candida albicans* strains used in this study.

Strain	Relevant Genotype	Source
SC5314	Clinical isolate from the London Mycological Reference Laboratory	[23]
BWP17	*ura3::imm434/ura3::imm434 his1::hisG/his1::hisG arg4::hisG/arg4::hisG*	[24]
Ume6-myc	As BWP17 except for *ume6::HIS1/UME6-9myc-NAT1*	In this study
Nrg1-myc	As BWP17 except for *nrg1::HIS1/NRG1-9myc-NAT1*	In this study
tetO-UME6	As BWP17 except for *ADH1/adh1::Ptet-UME6*	In this study
12–99	Fluconazole resistant strain	[25]
89	Caspofungin resistant strain	[26]
177	Caspofungin resistant strain	[26]

### 2.2. Antifungal Susceptibility Test

The minimum inhibitory concentrations (MICs) of 31 carbazole derivatives against *C. albicans* were determined using the broth microdilution method as described in the CLSI guidelines [22]. The tests were performed in 96-well flat-bottomed microtiter plates. The final concentration of the cell suspension in the RPMI 1640 medium was 2 × 10^4^
*C. albicans* SC5314 cells/mL, and the final concentration of the carbazole derivatives ranged between 0.125 and 64 µg/mL. All the wells were filled with RPMI 1640 to a final volume of 200 µL. The plates were incubated at 37 °C for 24 h. Growth inhibition was determined by measuring the optical density at 595 nm using a microplate reader. The MIC was defined as the lowest concentration of carbazole derivatives that inhibited the growth of cells by 95% compared to the control.

### 2.3. Hyphae Formation

YPD medium containing 10% FBS at 37 °C was used to test the effect of Molecule B and Molecule C on hyphal formation. *C. albicans* SC5314 (2 × 10^6^ cells/mL) was inoculated in hyphal formation-inducing media containing Molecule B and Molecule C at MICs and incubated at 37 °C with shaking for 2 h. The morphology of the cells was photographed using a Zeiss Observer.Z1 (Carl Zeiss, Jena, Germany).

### 2.4. Determination of the Minimum Inhibitory Concentrations of C. albicans Biofilm Formation

Inhibition of biofilm formation was investigated by seeding 2 × 10^4^
*C. albicans* SC5314 cells/mL in a 200 µL cell suspension into 96-well flat-bottom microtiter plates, adding Molecule B and Molecule C to final concentrations ranging between 0.125 and 64 µg/mL, and incubating at 37 °C for 24 h. Each well was washed with 200 µL of PBS three times to remove planktonic cells. A metabolic assay based on the reduction of XTT (sodium 3′-[1-(phenylaminocarbonyl)-3,4-tetrazolium]-bis (4-methoxy6-nitro) benzene sulfonic acid hydrate) was performed to determine the biofilm formation inhibitory concentrations. It referred to the lowest concentrations, where there was a 95% reduction in the XTT-colorimetric readings compared with the control. Colorimetric absorbance was measured at 495 nm using a microtiter plate reader.

### 2.5. Cell Cytotoxicity Test

The cytotoxicity of the carbazole derivatives was determined in HeLa cells. HeLa cells were maintained as previously described [27,28] in Dulbecco’s modified Eagle’s medium (DMEM, Invitrogen, Waltham, MA, USA) supplemented with 10% FBS (Invitrogen, Waltham, MA, USA) and grown at 37 °C in a humidified atmosphere of 5% CO_2_. Viability was measured using the MTS assay. Each well was inoculated with 104 HeLa cells and incubated at 37 °C for 24 h in 96-well flat-bottom tissue culture plates using DMEM containing 10% FBS. Carbazole derivatives were added at final concentrations ranging from 0.125 to 64 µg/mL and incubated at 37 °C for 16 h. The MTS solution with PMS was added, and the cells were incubated for 10 min. The absorbance at 490 nm was measured using a microplate reader.

### 2.6. Real-Time PCR

Real-time PCR was conducted as described previously [27,29,30,31]. Overnight cultured *C. albicans* cells were diluted in fresh YPD for 2 h at 30 °C to induce yeast growth. Overnight cultured *C. albicans* cells were diluted in fresh YPD supplemented with 10% FBS for 2 h at 37 °C to induce hyphal formation in the absence or presence of Molecule B and Molecule C at the final MICs. Total RNA samples of yeast or hyphal-formed cells were isolated using the bead beater (MP FastPrep-24, cooled on ice for 3 min each time, 12 times for 15 s.) and TRIzol reagent method. Total RNA was measured using the Nanodrop 1000 Spectrophotometer (Thermo Scientific, Rockford, IL, USA). Total RNA was treated with RQ1 DNase (Promega, Madison, WI, USA) and was reverse transcribed into cDNAs using oligo-d (T) and M-MLV reverse transcriptase (Promega, Madison, WI, USA). The cDNA preparation was diluted by 1/20 in Ultrapure distilled water (Invitrogen, Waltham, MA, USA) and stored at −20 °C until processing for real-time PCR. SensiFAST SYBR master mix (Meridian Bioscience, Cincinnati, OH, USA) was used to monitor the amplified products in real-time. Real-time PCR was performed with the specific primers listed in Appendix A using the LightCycler 480 (Roche, Manheim, Germany). Relative levels of target gene expression were calculated and normalized to the expression of *ACT1*.

### 2.7. Immunoblotting

Western blotting was performed as described previously [20,31,32], with slight modification. Briefly, yeast or hyphal formed cells were harvested and washed with ice-cold PBS. To prepare whole cell lysates, the cells were lysed with ice-cold buffer comprised of 50 mM Tris-HCl (pH 7.5), 150 mM NaCl, 5 mM EDTA, 10% glycerol, 0.2% Nonidet P-40, 1 mM dithiothreitol, protease inhibitors (1 mM phenylmethylsulfonyl fluoride and 1 μg/mL aprotinin, pepstatin A, and leupeptin), 0.1 mM sodium orthovanadate, 20 μM sodium glycerophosphate, 20 μM para-nitrophenyl phosphate, and 20 μM sodium fluoride. The cells were lysed using glass beads (Sigma-Aldrich, Burlington, MA, USA) in a Fast prep-24 device (MP Biomedicals, Santa Ana, CA, USA). The proteins were resolved on 10–12% SDS-PAGE and analyzed using immunoblotting on PVDF membranes according to standard procedures. For the detection of Ras1, Ume6-myc, Nrg1-myc, Cek1, Mkc1, phosphorylated Cek1, and phosphorylated Mkc1, the primary antibody (1:3000 dilution for Ras1 (05-516 Merck, Darmstadt, Germany) and 1:1000 dilution for c-Myc (c-Myc (9E10) Santa Cruz Biotechnology, Dallas, TX, USA), 1:1000 dilution for Cek1/Mkc1 (9102S Cell Signaling Technology, Danvers, MA, USA), and 1:1500 dilution for phospho-Cek1/phosphor-Mkc1 (4307T Cell Signaling Technology, Danvers, MA, USA) were used. For the loading control, β-actin antibody (ab8224 Abcam, Cambridge, UK) was used at 1:1000 dilution. The membranes were blocked using Tris-buffered saline containing 0.1% Tween, containing 5% skim milk. The BM Chemiluminescence Western Blotting Substrate kit was used to develop the blot.

### 2.8. Antifungal Activity Test in the Murine Candidiasis Model

Overnight-cultured *C. albicans* strains were pelleted and washed three times with 1 mL of sterile PBS. Next, 1.0 × 10^6^
*C. albicans* SC5314 cells were resuspended in 200 µL PBS and injected into 5-week-old BALB/c (female) mice via lateral tail veins. After the establishment of the murine model, mice (*n* = 5/group) were treated by tail vein injection with Molecule B and Molecule C (8 or 16 mg/kg) or PBS every day from the second day. To investigate the voluntary intake effect, the mice were treated with Molecule B and Molecule C (8 or 16 µg/mL) or normal water every day. The dead mice were dissected, the kidneys were harvested, weighed, washed with PBS, chopped, spread on YPD plates, and incubated at 30 °C for 2 days, then the CFUs were counted to analyze the fungal infection burden. The mice were monitored for 1 month, and Kaplan–Meier survival curves were plotted.

### 2.9. Ethics Statement

This study was performed in accordance with the guidelines of the Institutional Animal Care and Use Committee of Korea University. The protocol and experiments were approved by the Institutional Animal Care and Use Committee of Korea University. The permit number is KUIACUC-2019-0061. All mice used in the experiments were euthanized with minimal suffering.

## 3. Results

### 3.1. Identification of Carbazole Derivatives That Inhibit the Growth of Candida Albicans

To screen for carbazole derivatives that have growth inhibitory effects on *C. albicans*, the MICs of 31 carbazole derivatives were investigated. The MIC of Molecule B was 8 μg/mL, which showed the best effect among the 31 carbazole derivatives (Figure 1a and Appendix A, and Table 2). In addition, Molecule B was noncytotoxic in mammalian cells up to a concentration of 16 μg/mL (Figure 1b and Appendix A). The noncytotoxic concentration of Molecule B, 16 μg/mL, corresponds to twice its MIC, 8 μg/mL, so it was considered suitable for the experiment, and Molecule B was selected as one of the leading carbazole derivatives. Carbazole derivatives with a MIC of 16 μg/mL were Molecule C, Molecule C-M5, Molecule C-M6, Molecule C-M10, Molecule F, and Molecule N (Figure 1a). However, Molecule C-M5, Molecule C-M6, Molecule F, and Molecule N were excluded because their MICs showed cytotoxicity in mammalian cells (Figure 1b). Molecule C and Molecule C-M10 were noncytotoxic to mammalian cells even when their concentration was 32 μg/mL, which corresponds to a double MIC, 16 μg/mL (Figure 1b and Appendix A). Of the two, Molecule C was water-soluble; thus, it was considered advantageous for new drug development. Therefore, Molecule B and Molecule C were selected as the two leading carbazole derivatives in this study. Surprisingly, the MICs of Molecule B and Molecule C for fluconazole- and caspofungin-resistant *C. albicans* strains showed the same MIC as that of the reference strain (Table 2). Molecule B and Molecule C both showed the same MICs (4 µg/mL) in BY4741, a wild-type strain of *S. cerevisiae* (Appendix A). These results suggest that Molecule B and Molecule C have growth inhibitory activity against *C. albicans*, including strains that are resistant to existing drugs.

### 3.2. Inhibitory Effects of Molecule B and Molecule C on Morphogenesis and Biofilm Formation in Candida Albicans

In the hyphae inducing condition, hyphae were formed in 96% of cells in the control group; however, when treated with Molecule B and Molecule C, hyphae were formed in only 3% and 6% of cells, respectively (Figure 2a). The microscopy images showed the inhibitory effect of Molecule B and Molecule C on morphogenesis (Figure 2b). Biofilm can attach to host organs or medical devices and plays an important role in adhesion to and penetration of host tissues [33,34]. It was observed that biofilm formation was also inhibited by Molecule B and Molecule C at a minimum concentration of 1 μg/mL (Figure 2c). These results suggested that Molecule B and Molecule C can significantly inhibit pathogenicity by inhibiting the hyphal morphogenesis of *C. albicans.*

### 3.3. Virulence Diminishes in the Candidiasis Murine Model Treated with Molecule B and Molecule C via Oral Intake and Vein Injection

In addition to in vitro investigations, we conducted studies using a candidiasis mouse model to investigate whether Molecule B and Molecule C have antifungal properties against *C. albicans* in vivo. Both groups of control candidiasis mice died after 9 or 8 days (Figure 3a,c). On the other hand, the mice treated with Molecule B and Molecule C by tail vein injection showed higher survival rates (Figure 3a). The degrees of *Candida* infection of the kidneys, which are a major target of *C. albicans* [35], were also markedly lower in the groups administered Molecule B and Molecule C by tail vein injection (Figure 3b) than the control group. It was also observed that the survival rates of the mouse groups fed water containing Molecule B and Molecule C for autonomous intake were also very high (Figure 3c). In addition, the mice that drank water containing Molecule B and Molecule C showed a very low infection rate compared to the control group (Figure 3d). Through these results, it was confirmed that Molecule B and Molecule C were effective in inhibiting morphogenesis in vitro as well as in inhibiting pathogenicity in vivo.

### 3.4. Molecule B and Molecule C Inhibit the Morphogenesis of C. albicans by Regulating the Protein Levels of Nrg1 and Ume6

We investigated whether Molecule B and Molecule C decreased the level of HSGs associated with filamentous and invasive growth [13,36] in *C. albicans*. In *C. albicans* cells treated with Molecule B and Molecule C, the expression of *ECE1* encoding Ece1, a cytolytic peptide toxin essential for mucosal infection [37], was inhibited (Figure 4a). The expression of *ALS3*, encoding Als3, which plays an important role in cell adhesion and host surface adhesion [36,38], was also reduced (Figure 4a). In addition, the expression of *HWP1*, a cell-surface adhesion function in the hyphal cell wall [39,40], was markedly reduced (Figure 4a). Furthermore, it was confirmed that the expression of HGC1, which encodes a hypha-specific G1 cyclin-related protein essential for hyphal morphogenesis [41], was reduced (Figure 4a). Moreover, we continued to investigate two proteins, Nrg1 [13,14] and Ume6 [12], which act as negative feedback regulators at the transcriptional level under filament growth conditions, affecting the expression levels of HSGs. The Nrg1 protein level of *C. albicans* cells treated with Molecule B and Molecule C from 0 to 2 h was higher (Figure 4b) than the control. In addition, it was confirmed that the transcriptional level of *NRG1* was significantly increased in C. albicans cells treated with Molecule B and Molecule C (Figure 4c). On the other hand, Ume6 protein, which showed an increase with time in hyphae-induced *C. albicans* cells, was not detected at all time points in *C. albicans* cells treated with Molecule B and Molecule C (Figure 4d). In addition, it was observed that the mRNA expression level of *UME6* was significantly decreased in both the groups treated with Molecule B and Molecule C (Figure 4e).

We established a strain called tetO-UME6, in which one allele of UME6 was driven by a tetracycline-regulated promoter [42] to investigate whether the hyphal formation of *C. albicans* under constitutive *UME6* expression was inhibited by Molecule B and Molecule C treatment. Molecule B and Molecule C treatment under constitutive expression of *UME6* did not appear to be able to prevent the hyphal formation in *C. albicans* through direct Ume6 regulation (Appendix A). This result suggested that Molecule B and Molecule C treatments indirectly regulated the level of Ume6, which plays an important role in morphogenesis, thereby inhibiting *C. albicans* pathogenicity through the regulation of HSGs.

### 3.5. Molecule B and Molecule C Regulate the Ras1 and MAPK Pathways

We continued to investigate Ras signaling, which mediates hyphal growth induction in response to various environmental signals in *C. albicans* [7], to identify the upstream signaling pathway of Ume6. It was confirmed that *RAS1* mRNA expression was significantly inhibited in *C. albicans* cells treated with Molecule B and Molecule C (Figure 5a). It was also shown that the protein expression levels of Ras1 were significantly reduced in *C. albicans* cells treated with Molecule B and Molecule C (Figure 5b). To investigate the MAPK signaling pathway involved in the hyphal formation of *C. albicans* [43], mRNA and protein levels of MKC1 and CEK1 were investigated. The mRNA expression levels of *CEK1* and *MKC1* were decreased in *C. albicans* cells treated with Molecule B and Molecule C (Figure 5c,d). The protein levels of Cek1 and Mkc1 in *C. albicans* cells treated with Molecule B and Molecule C were not significantly different (Figure 5e). There was no difference in the change in the level of P-MKc1, an activated form of Mkc1 (Figure 5e). However, it was confirmed that the level of P-Cek1 was significantly reduced in *C. albicans* cells treated with Molecule B and Molecule C (Figure 5e). These results suggest that Ras1 is the key switch responsible for the antifungal effect of Molecule B and Molecule C through the Cek1 cascade activity in the MAPK pathway, which is well-known to be associated with the morphological transition in *C. albicans.*

## 4. Discussion

In this study, we first evaluated the inhibitory activity of 31 carbazole derivatives on the growth of *C. albicans* and their cytotoxicity to mammalian cells. Our results showed that Molecule B and Molecule C are appropriate agents for the growth inhibition of *C. albicans*. In addition, Molecule B and Molecule C were found to be nontoxic to mammalian cells at each tested MIC. We also confirmed the inhibitory effect of Molecule B and Molecule C on the growth of fluconazole- and caspofungin-resistant *C. albicans* strains. Many studies have suggested that the processes of biofilm formation [33] and morphogenesis [44] of *C. albicans* are closely related to pathogenicity. We found that Molecule B and Molecule C could inhibit the pathogenicity of *C. albicans* by inhibiting morphogenesis and biofilm formation. In addition, as a result of studying the survival rate of candidiasis mice by treatment with Molecule B and Molecule C in the in vivo experiment, it was confirmed that the pathogenicity of *C. albicans* was reduced. Surprisingly, not only candidiasis mice treated with Molecule B and Molecule C via tail vein injection but also mice given Molecule B and Molecule C dissolved in drinking water showed high survival rates. Both mice that drank water containing Molecule B and Molecule C and mice treated with Molecule B and Molecule C via tail vein injection had very low levels of *Candida* infection in the kidneys compared to controls. We predict that the results of this in vivo study were due to the inhibitory effect of Molecule B and Molecule C on the morphogenesis of *C. albicans*. Therefore, Molecule B and Molecule C could be effective candidates for the development of novel antifungal agents.

Fluconazole (azole) and caspofungin (echinocandin), which are commonly used as antifungal agents, target enzymes essential for cell wall ergosterol biosynthesis and the process of fungal cell wall synthesis, respectively. In order to control the growth of *C. albicans* strains resistant to these drugs, novel antifungal agents with different targets than those commonly used are needed. Our results showed that Molecule B and Molecule C regulate the activity of Ume6 and Nrg1, which are involved in the expression of HSGs, as well as disrupting the Ras1/MAPK pathway, which is mainly used for morphogenesis in *C. albicans*. In addition, it was confirmed that the MAPK pathway signaling components *STE11* (MAPKKK), *HST7* (MAPKK), and *CEK1* (MAPK) were inhibited at the mRNA level by Molecule B and Molecule C. The effect of inhibition of biofilm formation even at concentrations lower than the MIC can be attributed to the suppression of the expression of *ALS3* and *HWP1* by Molecule B and Molecule C. We believe that the inhibition of the expression of *ALS3* and *HWP1* by Molecule B and Molecule C at a step corresponding to the surface adhesion step in the biofilm formation process makes biofilm initiation no longer possible. Als3 and Hwp1 have also been reported to play an important role in biofilm formation [40,45]. Given that Molecule B and Molecule C caused a reduction in the *Candida* infection of the kidneys in the candidiasis mouse model, we can conclude that Molecule B and Molecule C suppressed the pathogenicity of *C. albicans* by inhibiting hyphal formation through the Ras1/MAPK signaling pathway.

The model illustrated in Figure 6 demonstrates how Molecule B and Molecule C regulate filamentous growth in *C. albicans*. Molecule B and Molecule C regulate the ‘*RAS1* and *CEK1’*-mediated MAPK pathway (*STE11-HST7-CEK1*) which is associated with the hyphal cell wall construction and invasive growth. In this study, we introduced Molecule B and Molecule C, which reduce the virulence of *C. albicans* by inhibiting morphogenesis via the Ras1/MAPK pathway. Our findings suggest that Molecule B and Molecule C are promising compounds for the development of new antifungal agents for the treatment of systemic candidiasis and possibly, for other fungal diseases.

## Figures and Tables

**Figure 1 jof-07-00688-f001:**
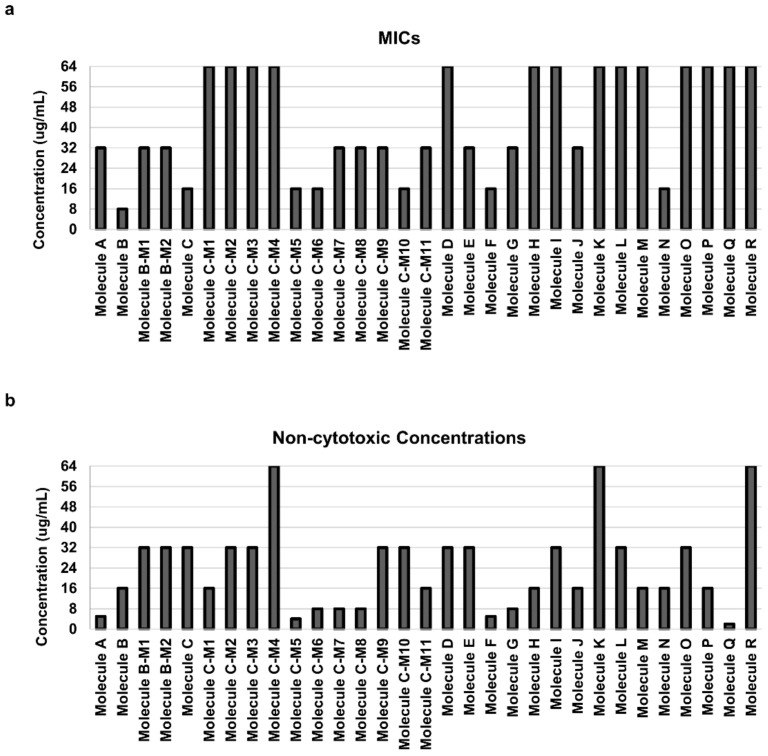
Cytotoxicity of carbazole derivatives in *C. albicans* and mammalian cells. (**a**) Minimum inhibitory concentrations (MICs) of 31 carbazole derivatives. MICs were defined as the lowest concentration of carbazole derivatives that inhibited the growth of the *C. albicans* SC5314 cells by 95% compared with the control. (**b**) Analysis of the cytotoxicity of carbazole derivatives on HeLa cells. The cytotoxicity was determined using the MTS assay. The noncytotoxic concentrations were defined as the highest concentration of carbazole derivatives that inhibited the growth of the cells by 5% compared with the control. The absorbance at 490 nm was measured with a microtiter plate reader. Each experiment was conducted in triplicate.

**Figure 2 jof-07-00688-f002:**
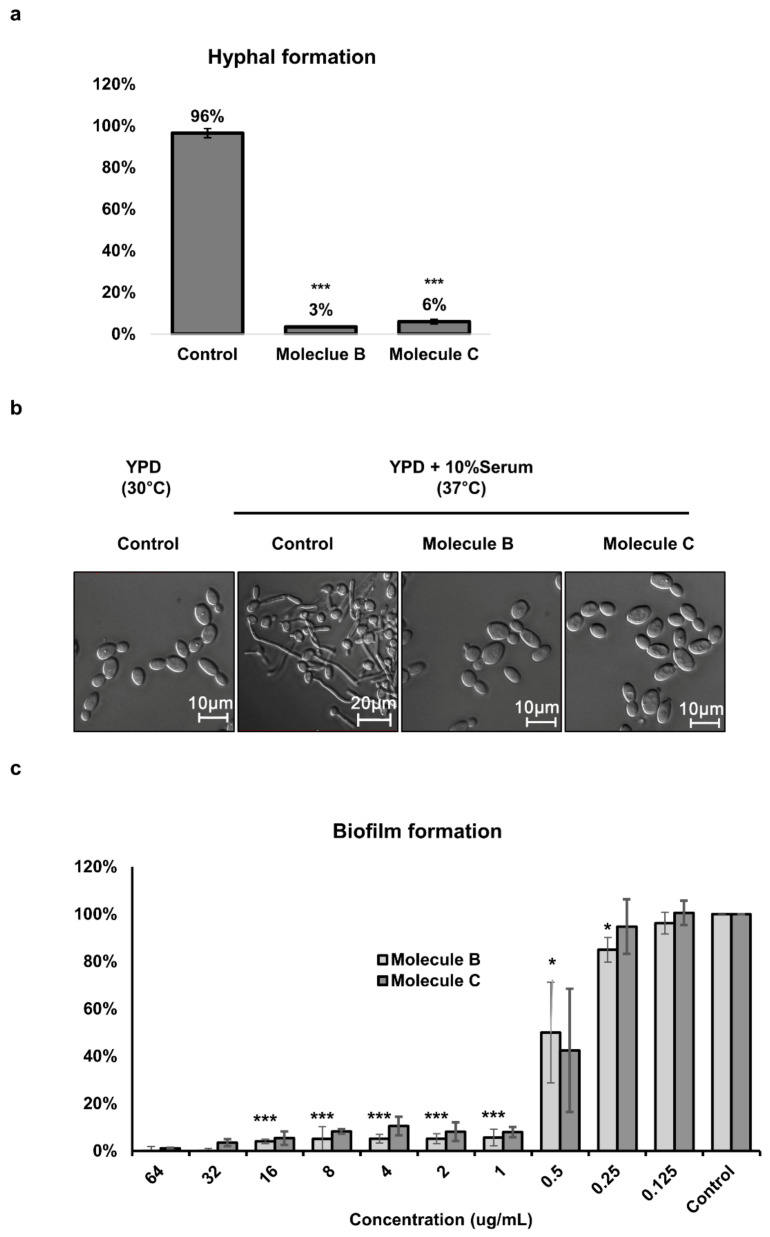
Inhibitory effect of Molecule B and Molecule C on morphogenesis and biofilm formation in *C. albicans.* (**a**) The inhibitory effects of Molecule B and Molecule C on the hyphal formation of *C. albicans* SC5314. To induce hyphae formation, YPD with 10% FBS was used and incubated for 2 h at 37 °C. Molecule B was treated at 8 μg/mL, and Molecule C was treated at 16 μg/mL. (**b**) A minimum of 300 *C. albicans* SC5314 cells were counted for each treatment to evaluate the morphogenesis inhibitory effect as a percentage. To observe the morphology change using a microscope, Molecule B and Molecule C were administered at each MIC. Scale bars represent 10 μm (yeast form) and 20 μm (hyphae). (**c**) The inhibitory effect of Molecule B and Molecule C on *C. albicans* biofilm formation. Biofilms were formed by inoculation 2 × 10^4^
*C. albicans* SC5314 cells/mL in 200 µL cell suspension into 96-well flat-bottomed microtiter plates adding Molecule B and Molecule C to final concentrations ranging between 0.125 and 64 µg/mL and incubating at 37 °C for 24 h. A metabolic assay based on the reduction of XTT was performed to determine the biofilm formation inhibitory concentrations. Colorimetric absorbance was measured at 495 nm in a microtiter plate reader. Each experiment was conducted in triplicate. The data represent the mean and standard deviation of three independent experiments. * *p* < 0.05, *** *p* < 0.001 (*t*-test).

**Figure 3 jof-07-00688-f003:**
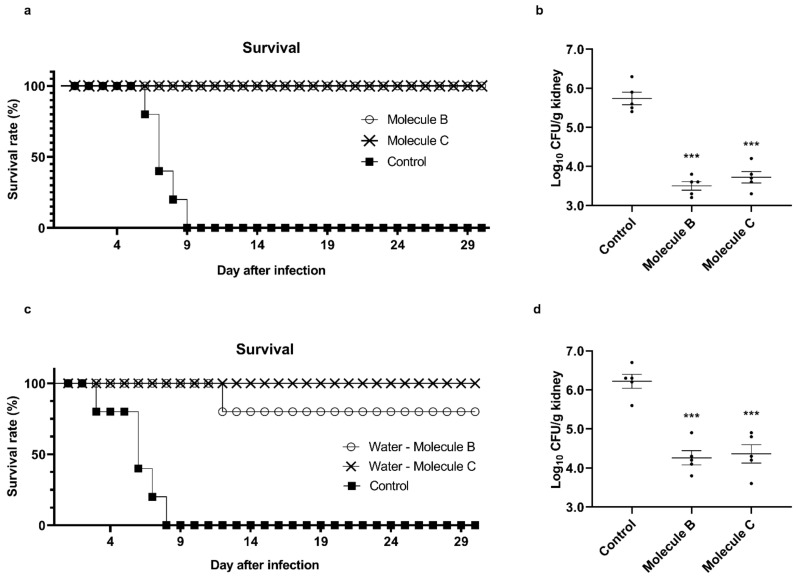
Attenuation of *C. albicans* pathogenicity by Molecule B and Molecule C treatment in the candidiasis murine model. The candidiasis mouse model was established by the inoculation of 5 × 10^6^ cells of *C. albicans* SC5314 through tail vein injection. (**a**) From the day after *C. albicans* infection to the 30th day, Molecule B and Molecule C were injected at 8 and 16 mg/kg, respectively, via tail vein injection, and the survival rate was measured every day. (**b**) The dead mice were dissected, the kidneys were harvested, weighed, washed with PBS, chopped, spread on YPD plates, and incubated at 30 °C for 2 days, and then the CFUs were counted to analyze the burden of fungal infection. Each Log_10_ CFU value is indicated by a dot, and the mean and standard error values are indicated by a horizontal line. (**c**) For 1 month from the day after *C. albicans* infection, Molecule B and Molecule C were diluted to 8 and 16 mg/L, respectively, in the drinking water of the mice, and the survival rate was measured every day after allowing the mice to ingest autonomously. (**d**) The dead mice were dissected, the kidneys were harvested, weighed, washed with PBS, chopped, spread on YPD plates, and incubated at 30 °C for 2 days, and the CFUs were counted to analyze the fungal infection burden. Each Log_10_ CFU value is indicated by a dot, and the mean and standard error values are indicated by a horizontal line. *** *p* < 0.001 (*t*-test).

**Figure 4 jof-07-00688-f004:**
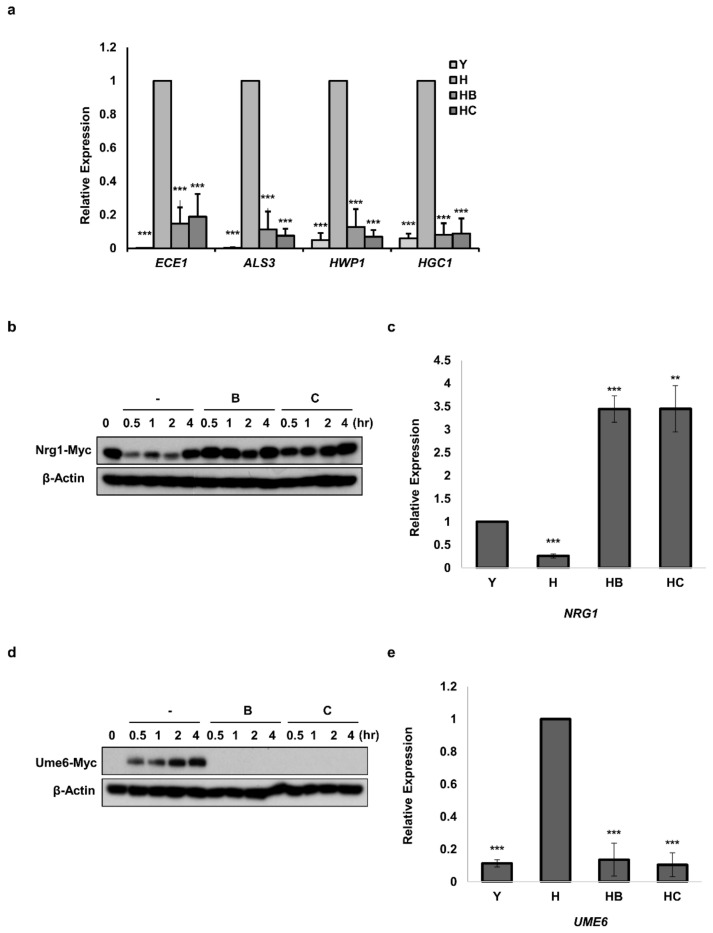
Effects of Molecule B and Molecule C on the expression of HSG, Ume6, and Nrg1 genes. (**a**) Realtime PCR for comparison of *HSGs* expression 2 h after hyphae induction. Used strain: BWP17. Y, yeast cells; H, hyphae induced cells; HB, hyphae induced cells with Molecule B treatment (8 μg/mL); HC, hyphae induced cells with Molecule C treatment (16 μg/mL). (**b**) Changes in the protein levels of Nrg1 after inducing hyphal formation (0–4 h). Used strain: Nrg1-myc. B, hyphae induced cells with Molecule B treatment of 8 μg/mL; C, hyphae induced cells with Molecule C treatment of 16 μg/mL. (**c**) Realtime PCR for comparison of *NRG1* expression 2 h after hyphae induction. Used strain: Nrg1-myc. Y, yeast cells; H, hyphae induced cells; HB, hyphae induced cells with Molecule B treatment (8 μg/mL); HC, hyphae induced cells with Molecule C treatment (16 μg/mL). (**d**) Changes in protein levels of Ume6 after inducing hyphal formation (0–4 h). Used strain: Ume6-myc. B, hyphae induced cells with Molecule B treatment (8 μg/mL); C, hyphae induced cells with Molecule C treatment (16 μg/mL). (**e**) Realtime PCR for the comparison of *UME6* expression 2 h after hyphal formation induction. Used strain: Ume6-myc. Y, yeast cell; H, hyphae induced cells; HB, hyphae induced cells with Molecule B treatment (8 μg/mL); HC, hyphae induced cells with Molecule C treatment (16 μg/mL). Western blotting was performed with an anti-myc antibody to detect Nrg1-myc and Ume6-myc. In western blotting, β-actin was used as a loading control by detection using the anti-β-actin antibody. Each experiment was conducted in triplicate. The data represent the mean and standard deviation of three independent experiments. ** *p* < 0.01, *** *p* < 0.001 (*t*-test).

**Figure 5 jof-07-00688-f005:**
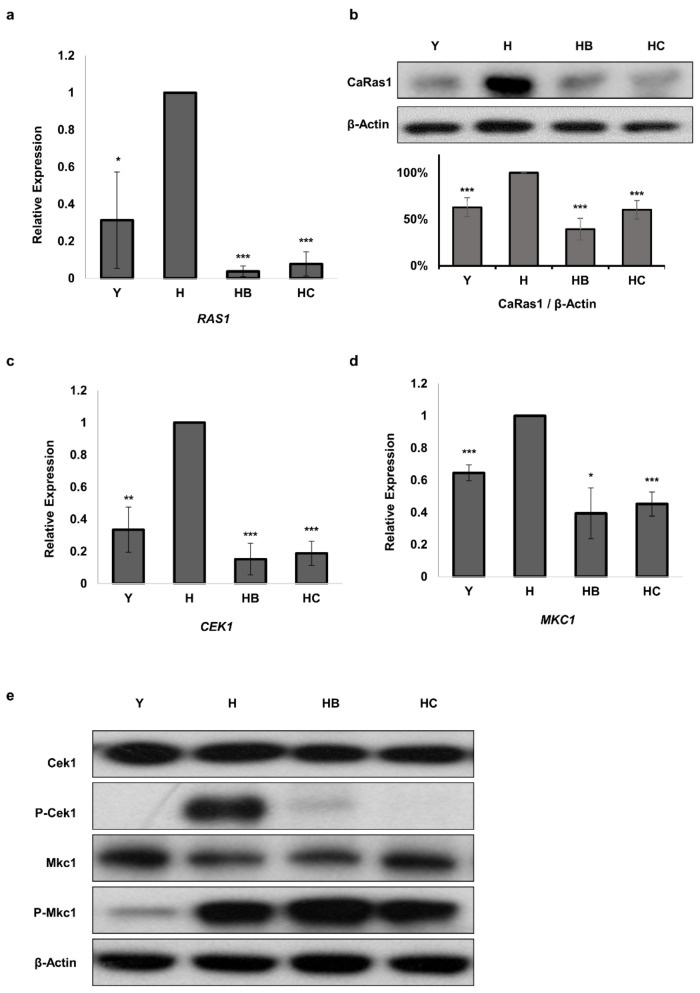
mRNA and protein expression levels of *RAS1* and mRNA expression levels of MAPK were reduced by Molecule B and Molecule C treatments. However, the level of P-Mkc1 did not decrease, while that of P-Cek1 did, with these treatments. The BWP17 *C. albicans* strain was used for western blotting and real-time PCR analyses. (**a**) Relative mRNA expression levels of *RAS1* were determined using real-time PCR. (**b**) In western blotting, CaRas1 was detected using the anti-Ras10 antibody, and β-actin was used as a loading control. The quantification graph by Image J software was prepared by normalizing the ratio of each CaRas1 blot to each β-actin loading control. The relative mRNA expression levels of *CEK1* (**c**), and MKC1 (**d**) were determined using real-time PCR. (**e**) Western blotting was performed by detecting Phospho-Mkc1 and Phospho-Cek1 using an anti-phospho-p44/42 antibody. Mkc1 and Cek1 were detected using the anti-p44/42 antibody, and β-actin was used as a loading control. Y, yeast cells; H, hyphae induced cells; HB, hyphae induced cells treated with 8 μg/mL Molecule B; HC, hyphae induced cells treated with 16 μg/mL Molecule C. Each experiment was conducted in triplicate. The data represent the mean and standard deviation of three independent experiments. * *p* < 0.05, ** *p* < 0.01, *** *p* < 0.001 (*t*-test).

**Figure 6 jof-07-00688-f006:**
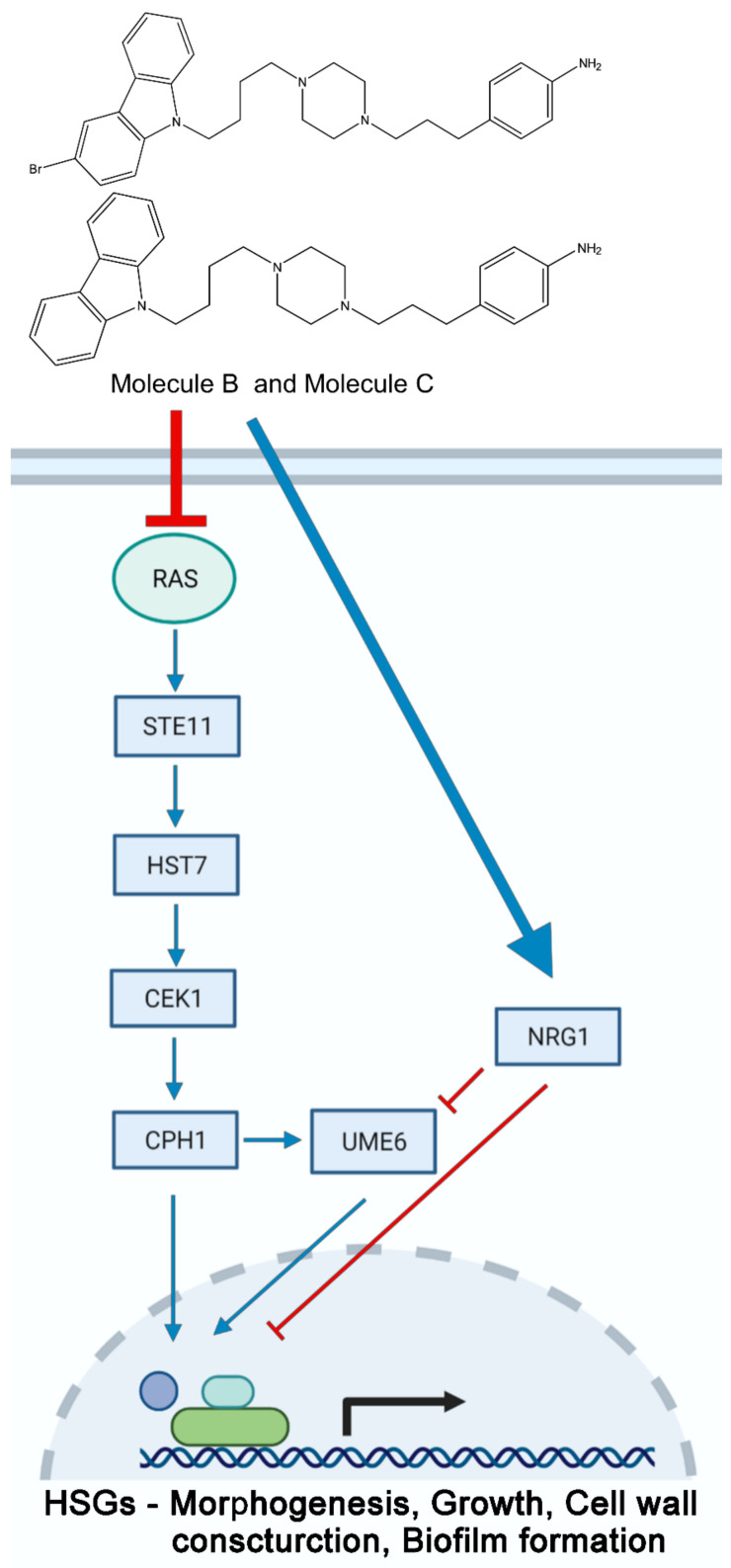
Putative model of virulence inhibition in *C. albicans* by Molecule B and Molecule C. The inhibitory effects of Molecule B and Molecule C on upstream activator signaling proteins are shown. Molecule B and Molecule C inhibit Ras1 and affect the PAK kinase Cst20, thus affecting the Cek1 MAP kinase cascade, resulting in a decrease in the levels of the transcription factor Cph1. The level of Ume6, which is regulated by the signal of Cph1, also decreases as the amount of Nrg1 increases.

**Table 2 jof-07-00688-t002:** MICs of Molecule B and Molecule C in *C. albicans*.

	Fluconazole	Caspofungin	Molecule B	Molecule C
SC5314 ^a^	1	1	8	16
12–99 ^b^	>128	1	8	16
89 ^c^	>64	4	8	16
177 ^c^	1	4	8	16

All units indicated are µg/mL. ^a^
*Candida albicans* reference strain. ^b^ Fluconazole resistant *Candida albicans* strain [25]. ^c^ Caspofungin resistant *Candida albicans* strain [26].

## Data Availability

The data presented in this study are available in the main article and Appendix A.

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
