# Peer review of "Development of Carbazole Derivatives Compounds against Candida albicans: Candidates to Prevent Hyphal Formation via the Ras1-MAPK Pathway"

_jof, 2021, doi:10.3390/jof7090688_

Round 1

Reviewer 1 Report

This study evaluated the effects of small molecules derivate from carbazole on C. albicans and candidiasis in mouse model. The results are interesting since two molecules were very promising. However, the text is poorly written. All the text needs a substantial revision.

1) Title:

-The title should include a word(s) that can specify the “novel drug candidates”. For example: “carbazole derivatives”.

2) Abstract:

-All the text is poorly written and needs substantial revision.

-Do not cite references in the abstract. Only summarize the proposal of this study and your principal findings.

-The following phrases are repetitive for an abstract: “The opportunistic human fungal pathogen Candida albicans has morphogenesis as a virulence factor. The morphogenesis of C. albicans is closely related to pathogenicity.”

-In the beginning of the abstract, the authors should write just one or two phrases for connecting the functions of the Ras1, MAPK pathway, Ume6 and NRG1. They are separated in the text, but seem to have closely functions.

-The objective of this study needs to be included more clearly in the text.

3) Introduction:

-In the phrase “Life-threatening systemic fungal infections occur mainly in immunocompetent and immunocompromised humans, such as AIDS patients, immunosuppressant-treated organ transplant patients, and recent COVID-19 patients”, only one reference is cited (Al-Hatmi et al. COVID-19 associated invasive candidiasis). References related to AIDS and organ transplant should be also included.

-Again, the text shows repetitive parts. For example, in the second paragraph: “Morphogenesis is one of the representative virulence factors of C. albicans” …… “Importantly, morphogenesis is linked to the virulence of C. albicans”.

-The third paragraph is very confusing. I suggest summarize it with focus on the pathways and genes that are investigated in this study.

-As described in the abstract, this study is focused on “novel small carbazole derivatives”, however in the introduction section little information is given about it: “Here, we describe novel small molecules that inhibit morphogenesis at elevated temperatures with serum incubation under hyphal induction conditions”. Add a paragraph in the introduction section for presenting these molecules. Describe the origin of these molecules, as well as a literature that show their biological properties known until now.

4) Materials and Methods

-In the phrase: “C. albicans strains, including FLC-resistant and echinocandin-resistant strains, primers, and small molecules used in this study are listed in Tables 1, 2, and 3, respectively”, the Table 1 shows results and not M&M. Therefore, Table 1 must be moved to the Results section.

-Replace “C. albicans strains, including FLC-resistant and echinocandin-resistant strains” by “C. albicans strains, including FLC-resistant and caspofungin-resistant strains”.

-In the Table 3, specify the antifungals of the echinocandins class that the 89 and 177 strains are resistant.

-Item “Antifungal susceptibility test”. In the phrase, “The minimum inhibitory concentrations (MICs) of the small molecules against C. albicans were determined…”. How many small molecules were evaluated? Write their codes or names.

-Rewrite the following phrase: “Their susceptibilities were determined according to the Clinical and Laboratory Standards Institute (CLSI) document M27-A3 [14] and RPMI 1640 (pH 7.0) was used as the liquid medium for diluting the drugs and strains. S. cerevisiae BY4741 cells were grown at 30°C in YPD medium, as previously described [15, 16]. All stock solutions of the chemicals were dissolved in sterile distilled water or methanol at a final concentration of 20 mg/mL. All stock solutions were stored at −20°C until use”. Why S. cerevisiae is cited here? Which antifungal compounds were tested here? The MIC values of the strains were determined in the present or previous study? As the authors declare in the beginning of M&M, FLC-resistant and echinocandin-resistant strains were selected for this study. So, I suppose that the MIC determination have been performed in previous study. This part is unclear.

-The item “Hyphae formation” should be placed after the item “Antifungal susceptibility test” since the MIC of small molecules was determined before the filamentation assay.

-Item “Real time PCR”: Describe how the yeasts and hyphal-formed cells were prepared and treated.

-Add more information about the methodologies used in the item Real time PCR.

5) Results

-The phrase “In our previous study [27], it was confirmed that the vacuole inhibitors have antifungal activity, and 32 relatively effective analogs of piperazinyl carbazole structure which was anticipated to possess vacuole inhibitory activity of bafilomycin A1 and compounds containing the carbazole structure [28-30] were also considered and screened. The active small molecules consist of carbazole, piperazine, and phenyl amine groups (Figure 1a)” is unclear here. This should be moved to introduction and/or M&M sections. I think that Figure 1a is not a result of this study. The results section should be limited to the results obtained in this study.

- The phrase “To screen for small molecules that have growth inhibitory effects on C. albicans, MICs of small molecules were investigated according to the Clinical and Laboratory Standards Institute (CLSI) document M27-A3 [14]” is not a result. It is M&M.

- In the paragraph of this phrase: “As shown in Table 1, the MICs of these small molecules were 16 μg/mL and 8 μg/mL, respectively, in SC5314, the wild type strain of C. albicans”, the authors only write “these molecules”. The Figure 1, there are 32 molecules. So, specify the molecules.

- The small molecules #2 and C#3 are also designed B and C. Please, use just one code in the text.

-Explain the reason for selecting the molecules #2 and C#3? In the Fig 1, there are molecules with lower toxicity and the same MIC in relation to C#3.

-Delete: “Growth inhibition is insufficient to prevent the virulence of C. albicans. Morphogenesis, a notorious virulence factor in C. albicans, is used as the standard for general testing of new drugs against C. albicans. To investigate the effect of inhibiting the morphogenesis of small molecules, hyphal formation was induced according to the presence or absence of small molecules. To induce hyphal morphogenesis, a YPD medium containing 10% FBS at 37°C was used and a YPD medium at 30°C incubation was used for the yeast form control. Small molecules B and C were used to treat hyphae formation at the concentration of each MIC. As a control group, the yeast form was shown in YPD medium at 30°C incubation, and the hyphal form was shown in YPD medium containing 10% FBS at 37 °C”. This is M&M and not results.

-Delete “Biofilm, a structured microbial community, is the final form of hyphal development in C. albicans, which can attach to host organs or medical devices and thus play an important role in adhesion and penetration into host tissues [4, 5]. To investigate whether small molecules B and C inhibit biofilm formation, biofilm was formed by densely attaching C. albicans to 96-well flat-bottomed microtiter plates for 24 h at 37°C. After the biofilm was densely formed, small molecules B and C were treated to a final concentration ranging from 0.125–64 μg/mL”. Again, this is M&M and not results.

-Specify the name of C. albicans strain in legends of all the figures.

-Again, most text of results section is M&M or literature, while the results are poorly described and unclear. All the section needs substantial improvement.

6) Discussion:

-Also need to be improved. Describe the discussion in a single text and not in separate items. The results should be better connected.

7) Changes needed in all the text:

-Use italic for gender and species of microorganisms

-English grammar must be revised

-Specify the name of “small molecules”

-Use superscript: 104 cells/mL

Author Response

Dear Reviewer 1,

Thank you for considering my article for publication in Journal of Fungi (jof-1310716). I am grateful to you for the valuable suggestions provided.

Again, thank you for giving us the opportunity to strengthen our manuscript with your valuable comments and queries. We have worked hard to incorporate your feedback and hope that these revisions persuade you to accept our submission.

Reviewer 2 Report

Dear Authors,

The manuscript ID: jof-1310716 entitled “Inhibition of Ras1-MAPK pathways for hypha formation by novel drug candidates in Candida albicans” written by Young Kwang Park, Jisoo Shin, Hee-Yoon Lee, Hag Dong Kim and Joon Kim is devoted to mechanism of action of novel molecules of carbazole derivatives.

Currently, the list of the commercially available antifungal agents, used for the treatment of infections caused by Candida spp. is limited to three major classes: polyenes, azoles and echinocandins. Moreover, drug resistance of yeast increases and the treatment of fungal infections is often ineffective. Therefore, is a need to search for new antifungals. The novel small molecules presented in these article showed different mechanisms of antifungal activity.

Introduction contains general data on the morphogenesis and antifungal resistance. The purpose of the work is concise and concrete. Appropriate methods were used to perform extensive research (assessment of hyphae formation, antifungal susceptibility, determination of MIC (minimum inhibitory concentrations) of Candida albicans biofilm formation, cell cytotoxicity test, Real-time PCR, immunoblotting, antifungal activity test in the murine candidiasis model). Results (identification of small molecules that inhibit the growth of C. albicans, inhibitory effects of tested molecules on morphogenesis and biofilm formation in C. albicans, results of virulence testing in a mouse model, mode of action on morphogenesis of C. albicans by regulating the protein levels of Nrg1 and Ume6) are documented, presented in the form of figures or table and right interpreted. Based on the results, adequate conclusions were drawn. The new carbazole derivatives exhibited noteworthy antifungal effect with specific mechanism of action. These molecules reduced virulence by inhibiting morphogenesis, which is a pathogenic factor of Candida via the Ras1/MAPK pathway. It is an interesting and original article.

However, I have some small suggestions in order to improve paper, which are the following:

  • Abstract – please do not cite the literature in the abstract;
  • C. albicans, Candida – please write in italics;
  • Introduction – please cite literature from position 1 (here is the 7-position);
  • The whole text should be harmonized in accordance with the instructions for authors;
  • Please also perform further tests on reference strains, because it would be more reliable

According to me, this manuscript is valuable and may be accepted for the publication in “Journal of Fungi”.

 With highest regards,

Author Response

Dear Reviewer 2,

Thank you for considering my article for publication in Journal of Fungi (jof-1310716). I am grateful to you for the valuable suggestions provided.

Again, thank you for giving us the opportunity to strengthen our manuscript with your valuable comments and queries. We have worked hard to incorporate your feedback and hope that these revisions persuade you to accept our submission.

Reviewer 3 Report

The manuscript describes the mode of action of a couple of molecules as inhibitors of C. albicans virulence. The subject is timely, the experiments were, overall, well conducted. The results are interesting and provide mechanistic details, but the manuscript is poorly written. The subject of novelty should also be clarified.

Major Issues:

English must be thoroughly revised by a native speaker. It’s quality is, sometimes, so bad that it compromises the ability to review the manuscript!

Carbazoles have been proposed before as antifungal, anti-Candida, molecules (e.g. a 2009 paper - https://pubmed.ncbi.nlm.nih.gov/19519387/). Please highlight the novelty, if any, of this manuscript.

Section 3.1 - What are “vacuole inhibitors”? What to they inhibit exactly? As it is, the first paragraph says that these compounds were already shown by these authors to have antifungal activity. If that’s the case, you must delete this section as it just reproduces published material…

Section 3.1 – Please provide structures for all the so-called “small molecules” in supplementary material, unless they were tested previously. If this is the case, just delete those results.

Figure 1b – Why was “Small molecule 3” selected? There’s a lot more with MIC 16…

Section 3.1 – I’m not sure I agree with the sentence “these molecules showed noncytotoxicity in mammalian cells at MIC”. MICs are actually quite close to the non-cytotoxic cut-off…

Figure 2c appears to suggest that the MIC concentration for molecules 2 and 3 is much lower in biofilm cells than in planktonic cells. This is very unusual to say the least, as biofilm cells are protected by the matrix and the remaining cell population! Please discuss these observations.

Throughout the manuscript, the use of “small molecules” is not appropriate. It could mean anything. ID the small molecules, and if you prefer use an abbreviation so that the reader knows what exactly is meant at each point by “small molecules” (e.g. subtitles 3.4 and 3.5, or even fig. 6)

Minor Issues:

Species names (e.g. Candida albicans) should be in italic. Please revise this in the whole manuscript.

Abstract: please rephrase the use of “small molecules” as the reader does not know what is meant. Even “small carbazole derivatives” is too vague. Please indicate the exact molecule(s) that you propose to be used as antifungal drugs.

Table 2 may be included as supplementary material.

Results:

Section 3.1 – “SC5314” is the “reference strain”, not “the wild-type strain” of C. albicans. Please rephrase.

Section 3.2 – The sentence “As a control group, the yeast form was shown in YPD medium at 30°C incubation, and the hyphal form was shown in YPD medium containing 10% FBS at 37 °C.” should be deleted as it just repeats the what was said two sentences back.

Section 3.2 – “Biofilm, a structured microbial community, is” NOT! “the final form of hyphal development in C. albicans,”. Please delete “, is the final form of hyphal development in C. albicans,”. Also delete “the final form of this morphogenesis”. This is an absurd idea!

Figure 5 Legend – Specify again what Y, H, HB, HC means, please.

Author Response

Dear Reviewer 3,

Thank you for considering my article for publication in Journal of Fungi (jof-1310716). I am grateful to you for the valuable suggestions provided.

Again, thank you for giving us the opportunity to strengthen our manuscript with your valuable comments and queries. We have worked hard to incorporate your feedback and hope that these revisions persuade you to accept our submission.

Round 2

Reviewer 1 Report

The authors performed a carefully review in the manuscript. The text was improved a lot in all the aspects: structure, grammar, organization and presentation of results. At moment, I have only some points for corrections or modifications:

-I suggest a modification in the title. The title “Development of carbazole derivatives against Ras1-MAPK pathways inhibiting Candida albicans hyphal formation” could be replaced by “Development of carbazole derivatives compounds against Candida albicans: candidates to prevent hyphal formation via the Ras1-MAPK pathway”.

-Introduction: The phrases “Fluconazole, an azole drug typically used as an antifungal agent, targets the enzyme essential for cell wall ergosterol biosynthesis. Other antifungal drugs, such as am-photericin B (polyene) and echinocandin target the fungal membrane and cell wall synthesis, respectively. In the ergosterol biosynthesis pathway, azoles inhibit ERG11…..commonly used drugs is emphasized” should be placed as a separate paragraph in the text, and not as part of the paragraph about Ras1-MAPK pathway.

-Correct the legend of Figure 1: “Cytotoxicity of carbazole derivatives in C. albicans and mammalian cells, and the chemical structures of Molecule B and Molecule C”, since chemical structures were removed.

-Table 2: add grids

-Delete the citations of Figures (with results) from the discussion section.

-Discussion section: the text is very clear now, but this section could be more explored, for example the results of in vivo study was not discussed. The study has excellent results that deserve been discussed.

Author Response

(The authors gave the same response as above.)

Reviewer 3 Report

All raised issues were adequately addressed.

Author Response

Dear Reviewer 3,

Thank you for considering my article for publication in Journal of Fungi (jof-1310716).